# Molecular Mechanisms of Anti-Estrogen Therapy Resistance and Novel Targeted Therapies

**DOI:** 10.3390/cancers14215206

**Published:** 2022-10-24

**Authors:** Rumeysa Ozyurt, Bulent Ozpolat

**Affiliations:** 1Department of Experimental Therapeutics, The University of Texas MD Anderson Cancer Center, Houston, TX 77030, USA; 2Houston Methodist Research Institute, Department of Nanomedicine, 6670 Bertner Ave, Houston, TX 77030, USA

**Keywords:** estrogen receptor-positive breast cancer, anti-estrogen therapy, resistance, SERM, SERD, aromatase inhibitors, targeted therapies, CDK4/6 inhibitors, RNAi

## Abstract

**Simple Summary:**

Estrogen receptor-positive (ER+) breast cancer is a notable cause of global death from breast cancer in premenopausal and postmenopausal women. Lethality is driven, amongst other things, by endocrine resistance, which is intrinsic or acquired in approximately 60% of patients after anti-estrogen therapy. Anti-estrogen therapy resistance, which has become a major clinical problem, results from high mutation rates of therapy targets, multiple subclonal diversity, dysregulation of downstream survival pathways, as well as alterations in the expression of non-coding RNAs (e.g., microRNA) that regulate the activity of signaling molecules. This review focuses on the current status of anti-estrogen therapy resistance and the key contributing mechanisms, as well as potential novel combination therapy strategies with anti-estrogen drugs to improve patient survival.

**Abstract:**

Breast cancer (BC) is the most commonly diagnosed cancer in women, constituting one-third of all cancers in women, and it is the second leading cause of cancer-related deaths in the United States. Anti-estrogen therapies, such as selective estrogen receptor modulators, significantly improve survival in estrogen receptor-positive (ER+) BC patients, which represents about 70% of cases. However, about 60% of patients inevitably experience intrinsic or acquired resistance to anti-estrogen therapies, representing a major clinical problem that leads to relapse, metastasis, and patient deaths. The resistance mechanisms involve mutations of the direct targets of anti-estrogen therapies, compensatory survival pathways, as well as alterations in the expression of non-coding RNAs (e.g., microRNA) that regulate the activity of survival and signaling pathways. Although cyclin-dependent kinase 4/6 and phosphatidylinositol 3-kinase (PI3K)/AKT/mammalian target of rapamycin (mTOR) inhibitors have significantly improved survival, the efficacy of these therapies alone and in combination with anti-estrogen therapy for advanced ER+ BC, are not curative in advanced and metastatic disease. Therefore, understanding the molecular mechanisms causing treatment resistance is critical for developing highly effective therapies and improving patient survival. This review focuses on the key mechanisms that contribute to anti-estrogen therapy resistance and potential new treatment strategies alone and in combination with anti-estrogen drugs to improve the survival of BC patients.

## 1. Introduction

Breast cancer (BC) is the most common malignancy in women around the world, a leading cause of cancer deaths in underdeveloped countries, and the second leading cause of cancer deaths in women in the United States. BC is characterized by different molecular, histological, and clinical profiles [1]. Estrogen receptor (ER), progesterone receptor (PR), and human epidermal growth factor receptor 2 (HER2) are three key molecular markers that are critical for prognostic classification and effective therapeutic selection in clinical BC practices [2,3]. BC is immunohistochemically classified into four major subtypes: (1) luminal A (ER+ and/or PR+ and HER2-; low Ki-67 level/low proliferation rate), (2) luminal B (ER+ and/or PR+ and HER2+; high Ki-67/high proliferation rate), (3) HER2-positive (ER-, PR-, and HER2+), and (4) basal/triple-negative (HER2-, ER-, and PR-) [4,5]. According to this classification, luminal A (ER+) BC is the most common subtype and represents about 70% of BC cases in women. Anti-estrogen therapy (e.g., tamoxifen, fulvestrant [Faslodex], letrozole) is the first choice for these patients after surgery, whereas chemotherapy is used as the main medical treatment modality for ER-/PR- patients, and HER2-targeted therapy (e.g., trastuzumab [Herceptin]) is used for HER2+ BC [4]. Triple-negative BC, which is characterized by the absence of ER, PR, and HER2, is a most aggressive subtype and, diagnosed in younger women, has the highest mortality rate among all BCs due to a lack of effective targeted therapeutics and low response rates for chemotherapeutics [6]. Overall, ER+ BC constitutes 70–80% of all BCs in premenopausal and postmenopausal women worldwide [4]. In such tumors, estrogen is the major oncogenic signaling factor and plays a major role in inducing cancer cell proliferation and tumor progression [4]. Although ER+ BC initially responds to anti-estrogen treatment, about 30% of ER+ patients will experience resistance to endocrine therapy, driving most BC deaths. In this review, we present the current status of anti-estrogen therapy resistance, the existing options, and further potential therapies that need to be considered to enhance strategies for overcoming anti-estrogen resistance.

## 2. The Estrogen/ERα Axis Is the Major Oncogenic Signaling Source in ER+ Breast Cancer Cases

ERs are members of a steroid receptor class in the nuclear receptor superfamily that act as transcription factors and are responsible for the regulation of certain target genes by binding to specific sides of target genes at the promoters on DNA [7]. ERs are predominantly located in the cytoplasm and nucleus but have also been associated with the plasma membrane [8,9]. Estrogen (17β-estradiol)-induced effects are functionally linked with two different nuclear receptors (Figure 1): ERα encoded by the ESR1 gene on chromosome 6 and ERβ encoded by the ESR2 gene on chromosome 14 [10,11]. Whereas both ERα and ERβ are expressed in human normal mammary tissues, ERβ is expressed at higher levels and acts as the primary receptor for the estrogen/ER axis [12,13,14]. ERβ is downregulated, but ERα is upregulated during carcinogenesis and drives BC cell growth, proliferation, and tumorigenesis [7,11,15].

Nuclear ERs consist of six functional domains [16]. The A/B domain at the N-terminal end in ERα and ERβ exhibits 18% homology [6,17]. This region of the receptors contains the transcriptional activation domain (AF-1) and explains the different effects of each ER isoform on target genes [6]. Phosphorylation of AF-1 leads to the activation of the ERs in a ligand-independent manner [18]. ERα and ERβ have 97% homology at the C-domain and contain the highly conserved DNA-binding domain and dimerization of the receptors [16,19]. ERα and ERβ exhibit 30% homology at the D-domain with the nuclear localization signal [16,20]. The E domain contains the ligand-binding domain (LBD), a co-activator/co-repressor interaction domain, a dimerization interface, and a ligand-dependent transcriptional activation domain (AF-2) [6,16,17]. The F-domain at the C-terminal end of the receptor is thought to protect the receptors against proteolysis and to modulate their interactions with co-activators and dimerization [16,20,21].

## 3. ERα Isoforms

Researchers identified four ERα isoforms with varying lengths (Figure 2): ERα66, ERα46, ERα36, and ERα30 [22]. ERα66 is a 66-kDa protein containing the typical functional domains of nuclear receptors in its structure, and that is used to determine ER+/ER- status in BC patients. ERα66 also plays a role in the genomic effects of estrogen [23]. ERα46, which lacks the F-domain, is involved in the nongenomic signaling of estrogen because it is mainly located at the plasma membrane [24]. ERα+ tumors with ERα46 expression are less malignant than other isoforms [24,25]. ERα36 overexpression leads to tamoxifen resistance and enhances metastatic potential. It lacks the AF-1 and AF-2 transcription activation domains but retains a truncated LBD. ERα36 is associated with proliferative, antiapoptotic, and metastatic effects and resistance to anti-estrogen therapy via a nongenomic pathway [26,27]. ER-α30 mediates the genomic signaling pathway, and its overexpression in triple-negative BC cells promotes malignant behaviors [28].

## 4. Signaling Pathways Mediated by the Interaction of Estrogen with ERα and ERβ

The first signaling pathway is initiated with the binding of estrogen to ERs in the nucleus. The ERs then bind to regulatory gene sequences in DNA, leading to the expression of a large number of target genes [29,30]. In addition, ERs can indirectly regulate the expression of receptor tyrosine kinases (RTKs) signaling pathways, including HER2, EGFR, and the phosphoinositide 3-kinase (PI3K) pathway that play crucial roles in the regulation of proliferation and differentiation by interacting with transcription factors without binding of ERs directly to DNA [11]. The second signaling pathway is mediated by membrane-associated ERα and ERβ and induces signaling molecules such as Src tyrosine kinase, mitogen-activated protein kinase (MAPK), PI3K/AKT, and protein kinase C, regulating the transcription of target genes by phosphorylating nuclear ERs [31,32,33]. Both nuclear and nonnuclear ERs interact with transmembrane growth factor receptor signaling pathways to support downstream signaling for tumor cell proliferation, survival, and treatment resistance. Therefore, an improved understanding of the molecular interactions of estrogen with other downstream signaling pathways involved in tumor progression is pivotal to developing new therapeutics targeting ER-mediated signaling pathways [31]. Figure 3 illustrates the ER signaling pathways.

## 5. Anti-Estrogen Therapies

Given the importance of ER signaling in breast tumor development, targeting ER has proven to provide significant clinical benefits and improved patient survival [4]. In fact, 70% of BC patients are ER+ and given anti-estrogen therapy as the standard first-line therapy. Anti-estrogen therapy has complementary effects after surgery, such as oophorectomy, adrenalectomy, and hypophysectomy. Anti-estrogen therapy for BC currently includes (1) selective ER modulators (SERMs), (2) selective ER downregulators (SERDs), (3) aromatase inhibitors (AIs), and (4) suppression of ovarian function, which is performed using gonadotropin-releasing hormone agonists, or a combination of two or more of these strategies [34]. Clinical applications of various anti-estrogen therapies are selected based on factors such as the patient’s tumor stage/metastatic status, reproductive age, menopausal status, and treatment history [35].

### 5.1. Selective ER Modulators (SERMs)

SERMs (e.g., tamoxifen, raloxifene) are anti-estrogen compounds that act as ERα antagonists, inhibiting estrogen-ER interaction by competing with estrogen and regulating the transcriptional activity of ERα [36]. Tamoxifen, the best-known SERM, is the preferred ER+ BC therapeutic, given orally to women before and/or after menopause; its active metabolite endoxifen may be an alternative treatment. Currently used SERMs comprise triphenylethylenes, consisting of tamoxifen and tamoxifen-like agents; benzothiophenes, including raloxifene and arzoxifene; phenylindoles, including basedoxifene and pipindoxifene; and tetrahydronaphthalenes, including lasofoxifene [35,37]. These agents antagonize estrogens in breast tissue and mimic estrogen in bone and endometrial tissues. The mechanism of these target-specific dual effects appears to be related to differences in the molecular and 3D structures of the co-activators and co-suppressors that affect the transcriptional activity of the ERs [4,38]. Tamoxifen is a prodrug that is converted to its active metabolites, including 4-hydroxytamoxifen (4-OHT) and endoxifen (4-hydroxy-*N*-desmethyl-tamoxifen), and 4-OHT and endoxifen is converted from *N*-desmethyl tamoxifen by cytochrome p450 enzymes [39,40]. 4-OHT has a higher affinity for ERα than tamoxifen [41,42], whereas endoxifen binds to ERs with a much higher affinity than 4-OHT. [43,44]. Endoxifen has exhibited antitumorigenic effects but also toxicity when tested in a phase 1 clinical trial [45]. Raloxifene has estrogen-agonist effects in bone and the cardiovascular system and estrogen-antagonist effects in the uterus and mammary glands. The active tamoxifen metabolite 4-OHT has a similar binding affinity for ERα and ERβ, whereas raloxifene has a much stronger affinity for ERα than ERβ [46]. In MCF-7 ER+ BC cells, the antigrowth effect of raloxifene was found to be stronger than that of 4-OHT [47,48].

### 5.2. Selective ER Downregulators (SERDs)

SERDs (e.g., fulvestrant) have antagonistic effects on ERα and ERβ. Fulvestrant has been approved by the U.S. Food and Drug Administration (FDA) for more than 10 years for the treatment of metastatic ER+ BC in postmenopausal women after tumor progression and after receiving anti-estrogen therapeutics such as tamoxifen. The use of fulvestrant leads to ERα downregulation by inducing its degradation [49]. When SERDs bind to ERs, they inhibit the activation of co-activators as well as disrupt the dimerization and nuclear localization of the ERs [50]. SERDs have greater ER-antagonistic effects than do SERMs and have better growth-inhibitory effects, even on tamoxifen-resistant BC cells [51]. SERDs exhibit improved clinical activity in treating ER+ BC in both early-stage and more advanced drug-resistant cases, whereas their intramuscular route of administration and low bioavailability are major clinical limitations [52,53].

### 5.3. Aromatase Inhibitors (AIs)

The aromatase (estrogen synthase) enzyme CYP19 catalyzes the final and rate-limiting step in the biosynthesis of estrogen. Thus, inhibitors of this enzyme are effectively used as targeted therapy for ER+ BC [54,55]. The FDA-approved agents letrozole, anastrozole, and exemestane are reversible AIs, with exemestane also used as an irreversible inhibitor aromatase. Unlike tamoxifen, AIs do not have a significant effect on the overall survival of ER+ BC [56,57]. Moreover, neither AIs nor fulvestrant has yet to replace tamoxifen as the primary anti-estrogen therapeutic for ER+ BC, as they cause toxic effects in premenopausal women [54]. More recently, it has been identified that the HER2-enriched subtype drives aromatase inhibitor resistance and increases the risk of relapse in early ER+/HER2+ breast cancer [55]. AIs now represent the standard treatment in postmenopausal patients and are effective as initial therapy for metastatic or locally advanced ER+ BC that has progressed after treatment with tamoxifen.

## 6. Mechanisms of Resistance to Anti-Estrogen Therapies

Anti-estrogen therapy has significantly improved survival of ER+ BC over the past three decades, but resistance to it is a major challenge in the treatment of ER+ BC and remains one of the main causes of poor prognosis and patient deaths. Studies have identified various mechanisms of resistance to anti-estrogen therapies. Most of the studies focused on the structure, activation, and complex functions of ERα as well as the interplay of the estrogen signaling network with other cellular pathways. The mechanisms causing resistance to anti-estrogen therapies are complex and originate from various molecular alterations and cellular factors, including loss of ERα expression, ERα mutations, changes in the downstream actions of ER target genes, loss of PR, increased tamoxifen metabolism and efflux via a multidrug resistance P-glycoprotein drug pump, overexpression of microsomal anti-estrogen-binding site proteins that bind to tamoxifen, cell-cycle regulators, microRNA (miRNA) dysregulation and increased activity of receptor tyrosine kinases (RTKs), growth factor pathways, and other signaling networks and cancer stem-like cells (Figure 4) [58,59].

### 6.1. ERα Mutations

In several recent studies, ERα mutations were detected in about 20% of metastatic ER+ BC patients receiving anti-estrogen therapies containing AIs [60,61,62]. Although ERα mutations are very rare in primary BCs, the frequency of these mutations increases with anti-estrogen therapy. Mutations of ERα concentrated in the LBD lead to ligand-independent ER activity and tumor growth, the development of tumor resistance to tamoxifen and even fulvestrant, and increased metastatic potential [63]. Whereas *ESR1* is rarely mutated in primary tumors, it is mutated in 20–55% of metastatic breast tumors, and these mutations are located in the LBD, leading to estrogen-independence and reduced response to targeted treatments. Recently, several reports have described point mutations of ERα in the LBD in metastatic ER+ BC cases [60,64,65,66]. These studies of a total of 187 metastatic ER+ tumors from patients receiving anti-estrogen therapy identified 39 cases with ER-LBD mutated. However, the prevalence of these mutations varied from 14% to 54% in different studies. In addition, Jeselsohn et al. [61] showed a correlation between the prevalence of LBD mutations and the number of anti-estrogen treatment lines. Detection of mutations of LBD in tumor specimens from different sites, including lymph nodes, skin, lung, and liver, suggests that these mutations do not have specific organotropism [60,61,62,64]. Most of the LBD mutations in these studies were missense mutations. Phosphorylation of the mutated/substituted residues of LBD is the most important factor in controlling the agonist status of, conformational changes in, and protein stability of the LBD of the ER. Thus, mutations of these residues lead to conformational changes in the ring connecting helix 11 and helix 12 in the LBD [65,66,67]. Moreover, the ER mutants Y537 and D538 are phosphorylated on S118 in an estrogen-independent manner by TFIIH kinase and cyclin-dependent kinase (CDK) 7. This may be a cause of the anti-estrogen resistance of BC by potentiating the transcriptional activity of mutant ER-derived cancer cells [66]. In addition, three double mutants (Y537N/D538G, S436P/D538G, and S436P/Y537N) have been detected within the same tumor [61,63,64].

### 6.2. Loss of ERα Expression

Because anti-estrogen therapy targets ERα, which acts as a growth factor and is required for the proliferation of breast tumor cells, as expected, researchers have found loss of ERα expression to be an important factor in the acquisition of anti-estrogen therapy resistance. Authors reported that loss of ERα expression contributed to the acquisition of endocrine resistance in about 20% of patients with ER+ BC [68]. Moreover, increased numbers of ERα- cells in BC tissue are associated with metastatic recurrence [69]. CD146 (also known as melanoma cell adhesion molecule or MUC18) is highly upregulated in tamoxifen-resistant MCF-7 cells and associated with reduced ERα expression, and silencing of CD146 in tamoxifen-resistant cell lines has reversed tamoxifen resistance. Furthermore, in a subset of BC cell lines, CD146 expression suppressed ER expression, and CD146 overexpression induced Akt activation, which is one of the mechanisms that contribute to tamoxifen resistance [70]. Elevated CD146 expression was found in 8 of 317 ER+ BC patients, and its expression is significantly associated with poor prognosis for ERα+ BC treated with tamoxifen. CD146 expression is associated with high tumor grade and ER-, PR-, and high–Ki-67 tumors. Other studies with large ER+ BC patient numbers demonstrated that with low or no CD146 expression was responsive to treatment with tamoxifen [71]. Inhibition of HDAC3 has significantly reduced the viability of ERα-deficient endocrine-resistant BC cells, and the ERα/caspase 7/HDAC3 axis promotes the proliferation of endocrine-resistant BC cells that exhibit ERα loss [72]. Another study demonstrated that BC cells in lymph node metastases had lower levels of ERα expression than those in the primary tumors. More importantly, ERα loss in lymphatic metastases correlated positively with clinical stage and lymph node metastasis [73]. However, how endocrine-resistant BC cells retain their proliferative advantage after the loss of ERα expression remains unknown.

### 6.3. ERα36 Overexpression

ERα36 protein is detected in about 40% of ER+ cases and 50–100% of ER- TNBC cases. In ER+ MCF-7 BC cells, ERα36 overexpression leads to tamoxifen resistance, and tamoxifen acts as an ERα36 agonist, inducing proliferation, migration, and invasion of breast cancer cells. Thus, tamoxifen-based treatment leads to poor outcomes of highly ERα36-expressing ER+ BC [22]. Studies have suggested that elevated ERα36 expression transforms estrogen-dependent mitogenic signaling to estrogen-independent signaling, promoting the progression of ER+ BC [74].

### 6.4. Loss of Progesterone Receptor (PR) Expression

In ER+ BC cases, PR is often used as a positive prognostic factor, but the role of PR signaling in these cancers is still unclear [75]. Regarding advanced BC, about 65% of all ER+/PR+ BCs respond well to anti-estrogen therapy [76], whereas ER+/PR- BC patients have a markedly worse prognosis [77,78]. ER is the transcription factor for PR, leading to PR expression [79]. PR is not only an ERα-induced gene target but also an ERα-related protein that modulates its behavior by controlling chromatin binding and transcriptional activity, resulting in important implications for prognosis and therapeutic interventions [80]. A recent study showed that loss of PR expression has a significant impact on survival in ER+ BC patients and that this loss significantly affects downstream signaling by involving several PI3K pathway-dependent kinases [79]. In addition, in patients with metastatic ER+ cancers, PR expression leads to an enhanced response to tamoxifen and is associated with longer overall survival than PR- BC patients. On the other hand, PR negativity in ERα+ BC patients is associated with better response rates for neoadjuvant chemotherapy than in ER+/PR+ BC patients [81,82]. Therefore, PR status may be the determining factor in predicting response to neoadjuvant chemotherapy in ERα+ BC patients [83]. Lack or loss of PR expression in ERα+ BC cases may be due to various potential mechanisms, but the exact molecular, pathological, and clinical heterogeneity of PR remains poorly understood.

### 6.5. Increased Tamoxifen Metabolism

Tamoxifen is metabolized to the active metabolites 4-OHT and endoxifen, the latter of which is the most potent metabolite of tamoxifen, and binds to ERs with much higher affinity than tamoxifen [43,44]. The metabolite concentration in plasma determines the drug’s efficacy, and endoxifen is considered the most important metabolite in determining the clinical efficacy of the parent drug, as endoxifen has a higher steady-state plasma concentration than 4-OHT [84]. CYP2D6 is the main enzyme in the metabolism of tamoxifen to 4-OHT and endoxifen and is one of the most important enzymes in determining plasma endoxifen concentrations [85]. However, many other enzymes, such as CYP3A4, CYP2C9, CYP2B6, and CYP2C19, are involved in the metabolism of tamoxifen [86]. Recently, authors reported that CYP2C19 may be the most important risk factor for tamoxifen resistance or response [87,88]. The relationship among CYP2D6 activity, endoxifen level, and anti-estrogenic activity of endoxifen was recently described by Wu et al. [89]. CYP2D6 is a highly polymorphic enzyme, and blood concentrations of endoxifen have been shown to correlate significantly with the CYP2D6 phenotype [90,91]. In addition, Goetz et al. [92] reported that patients with dysregulation of CYP2D6 metabolism had a twofold increase in the risk of BC relapse when given tamoxifen as an adjuvant.

### 6.6. Loss of the Cell-Cycle Regulator Retinoblastoma Protein

Loss of function of the retinoblastoma (Rb) protein is reported to be an important reason for the failure of anti-estrogen therapies for ER+ BCs. For example, in anti-estrogen therapy-resistant MCF-7 BC tumors, cyclin D1 expression and RB phosphorylation were preserved despite ER inhibition [93]. The Rb protein (pRb) is the main downstream target of CDK4/6 inhibition. Researchers have suggested that pRb activity must be intact for ER+ BC cancer cells to be sensitive to treatment with CDK4/6 inhibitors [94]. Indeed, preclinical and clinical studies have shown that loss of Rb function is associated with resistance of ER+ BC to CDK4/6 inhibitors [95,96,97,98,99]. A high expression of Rb1, as well as an upregulation of cyclin D1, were detected in cells sensitive to treatment with the CDK4/6 inhibitor palbociclib [95]. Furthermore, a study using an animal model of basal-like breast cancer showed that Rb-deficient tumors in C3-Tag mice were resistant to palbociclib [100]. Recent clinical data from genetic analysis of ER+ BC patients given CDK4/6 inhibitors revealed multiple genetic alterations, including biallelic inactivation of Rb1, as potential biomarkers of CDK4/6 inhibitor resistance [98]. However, despite its remarkable effect on CDK4/6 resistance, some studies do not support a relationship of Rb expression with response to CDK4/6 inhibition. More preclinical and clinical studies are needed to elucidate this issue.

### 6.7. Increased Activity of Receptor Tyrosine Kinases (RTKs)

Increased RTK levels enhance the aggressive potential of BC and reduce patient survival, resulting in poor prognosis [101]. The interaction of ERα with growth factor RTK signaling is reciprocal, whereby ERα promotes the transcription of RTKs, including epidermal growth factor receptor (EGFR), HER2, insulin-like growth factor-1 receptor (IGF-1R), fibroblast-like growth factor receptor 1 (FGFR1), and FGFR3 [102]. The MAPK, Janus kinase/signaling transducer and activator of transcription, and PI3K/Akt signaling pathways are important intracellular pathways activated by RTKs [103,104,105,106]. Overexpression of RTKs such as EGFR, HER2, and IGF-1R promotes the downregulation of ERα and leads to resistance of ER+ BC to anti-estrogen therapy through activation of the PI3K/Akt and MAPK pathways [107,108]. The EGFR/MAPK axis enhances phosphorylation of the AF-1 region of the ERs to induce ligand-independent activation of ER signaling [108,109]. Activation of EGFR/ErbB2 signaling in tamoxifen-resistant ER+ BC cells supports an aggressive and malignant stem cell phenotype of these cells [110,111]. IGF-1R plays a key role in the development of aggressive BC through Ras/Raf/MAPK kinase 1/2/extracellular signal-regulated kinase 1/2 and PI3K/AKT/mammalian target of rapamycin (mTOR) signaling. Activation of this receptor causes ERα levels to increase in extranuclear domains and leads to ligand-independent activation of the ER, which further stimulates Ras/Raf/MAPK kinase 1/2/extracellular signal-regulated kinase 1/2 and PI3K/AKT/mTOR signaling, resulting in endocrine resistance [112]. PI3K signaling interacts directly and indirectly with ERα. Phosphorylation of ERα at Ser167 by AKT or p70S6K confers tamoxifen resistance [113,114]. In addition, PI3K and Ras promote c-Jun phosphorylation [115,116]. c-Jun interacts with c-Fos to form the AP-1 complex, which cooperates with ER transcription [117,118]. Overactivation of FGFR3 reduces the sensitivity of ER+ BC to tamoxifen via Ras/Raf/MAPK kinase 1/2/extracellular signal-regulated kinase 1/2 and PI3K/AKT/mTOR signaling [119]. In addition, FGFR1 contributes to anti-estrogen resistance by triggering cyclin D1 in ER+ BC cells [120].

### 6.8. Breast Cancer Stem Cells (BCSCs)

Breast cancer stem cells (BCSC) play a supportive role in anti-estrogen therapy resistance in ER+ BC. BCSCs mediate the recurrence to anti-estrogen therapy, which is consistent with the consensus that cancer stem cells (CSCs) are a driver of cancer evolution and resistance to treatments [121,122]. The contributions of BCSCs to anti-estrogen resistance regulated by multiple mechanisms offer a therapeutic strategy to counter the resistance. In this section, we outline the evidence supporting that BCSC is a cause of anti-estrogen therapy resistance and discuss strategies to reduce anti-estrogen therapy resistance by targeting BCSCs.

Mammary stem cells (MaSCs) with the ability to regenerate mammary glands localized in the mammary gland epithelial compartment have been demonstrated by the transplantation approach [123] and lineage tracing experiments [124]. Although markers including Lin^−^CD29^hi^CD24^+^ (Lin^−^: lineage negative) [125], Lgr5^+^Tspan8^hi^ [126], Bcl11b^+^ [127], CD49f^hi^EpCAM^−/low^ [128], and ALDH1^+^ [129] have been reported in these cells, the identity of MaSCs is not entirely clear. BCSCs also have heterogeneous marker expression. Following the initial identification of BCSCs as CD44^+^CD24^−/low^ESA^+^Lin^−^ (ESA: epithelial-specific antigen) [130], ALDH1^+^ [129], CD49f^+^ [131], and CD133^+^ [132] were also shown to be expressed in BCSCs. CD133^+^ and CD49f^+^ BCSCs are enriched in TNBC and are associated with resistance to chemotherapy [131,133]. CD44^+^CD24^−/low^ and ALDH1^+^ BCSC populations have been associated with adverse traits and poor prognosis in ER+ BCs; ALDH1+ BCSCs are particularly associated with early relapse following anti-estrogen therapy [134,135].

Cumulative evidence reveals a positive contribution of BCSCs to the development of resistance to anti-estrogen therapy. BCSCs were enriched in established tamoxifen-resistant MCF7 cell lines (MCF7-TamR) compared with parental MCF7 cells. The rate of CD44^+^CD24^−/low^ cells in MCF7-TamR was elevated [121,136]; increase in ALDH1 [122], and CD133 [137] were reported in MCF7-TamR compared to parental MCF7 cells. 4-OH-tamoxifen treatment increased mammosphere forming capacity of estrogen-dependent LM05-E and MCF-7 cells. In addition, treatment with tamoxifen of mice carrying the M05 tumor (estrogen-dependent and sensitive to tamoxifen) increased the cells capable of forming mammospheres and CD29^hi^CD24^low^ BCSC population [138]. ER+ human breast tumors from patients treated with AI letrozol had higher CD44^+^CD24^−/low^ BCSCs than the untreated tumors [139]. It has been reported that short-term treatment with anti-estrogen therapies increases JAG1-NOTCH4-regulated ALDH1^+^ BCSCs in patient-derived ER+ BC tumors [140]. Interestingly, BCSC enrichment occurs at a very early stage of anti-estrogen therapy [138], and long-term therapy is required to reach a stable resistance state [141]. Targeting NOTCH4 antagonizes the increase in NOTCH and BCSC activity induced by tamoxifen or fulvestrant. Moreover, inhibition of NOTCH4 in tumors resistant to tamoxifen decreased BCSC activity. Thus, this study indicates that the combination of endocrine therapy with anti-JAG1-NOTCH4 overcomes resistance in ER+ BC [140]. Preclinical and clinical studies revealed that the NOTCH inhibitor (γ-secretase inhibitors) reduces stem cell markers in tumors and increases the efficacy of chemotherapy in breast cancer patients, but further studies are needed to confirm whether this effect applies to those receiving anti-estrogen therapy [142]. Death-associated factor 6 (DAXX), the negative regulator of NOTCH, suppresses the expression of genes such as SOX2, which inhibits BCSC proliferation following anti-estrogen therapy [143]. Thus, agents that stabilize DAXX show the potential to overcome endocrine resistance when used in combination with anti-estrogen therapy. In addition, RFS was significantly worse for patients receiving anti-estrogen therapy who had low DAXX levels [144]. These data demonstrate the necessity of using DAXX with appropriate therapeutic agents to overcome anti-estrogen therapy resistance.

The mechanisms underlying BCSC enrichment contributing to anti-estrogen therapy resistance can be classified as ER, growth factor receptors/PI3K-AKT-mTOR, NOTCH, Wnt, Hippo-YAP/TAZ, and microenvironmental effects. ESR1 gene mutations in at least Y537S, Y537N, and D598G contribute to anti-estrogen resistance in part via BCSC enrichment [145]. The ER co-activator AIB1/NCOA3/SRC3 (steroid co-activator 3) supports anti-estrogen therapy resistance via BCSC enrichment [146,147]. A profile study of CD44^+^CD24^−/low^ BCSCs isolated from primary tumors showed upregulation of the PI3K pathway, EGFR, other growth factors, Wnt and NOTCH pathways, and stemness genes [148]. In addition, mutations leading to activation of the PI3K-AKT-mTOR pathway were detected in BCSCs isolated from patient tumors [149]. Moreover, overexpression of the PI3K-AKT-mTOR pathway has been detected in BCSCs associated with tamoxifen-resistant tumors [150]. SOX2 has been reported to increase resistance to tamoxifen by stimulating BCSCs, a process partially mediated by SOX2-mediated Wnt signaling [151]. Wnt signaling works downstream of SOX9 to facilitate BCSCs, thereby inducing anti-estrogen therapy resistance [152]. TAZ is required for self-renewal of CD44^+^CD24^−/low^ BCSCs [153]. High levels of YAP/TAZ expression are associated with metastasis, poor survival, and resistance to treatments in patients with breast cancer [154]. Furthermore, YAP/TAZ transcriptional coactivators cross-talk with Wnt, Hedgehog, and NOTCH signaling pathways, in doing so contributing to YAP/TAZ in maintaining the self-renewal potential of BCSC [155]. The tumor microenvironment contributes to anti-estrogen therapy resistance in part by regulating BSCSs. Extracellular vesicles derived from cancer-associated fibroblasts induce anti-estrogen resistance by regulating BCSCs [156]. Inflammatory cytokines including IL-8, IL-6, IL-33 released from the microenvironment contribute to the self-renewal of BCSCs in the anti-estrogen resistance [157,158,159,160].

The contribution of BCSCs to anti-estrogen therapy resistance has been extensively studied. Currently, information has accumulated on the mechanisms that regulate BCSCs following anti-estrogen therapy, but translating this information to the clinic is a challenge.

### 6.9. Dysregulation of microRNA (miRNA) Expression

MiRNAs are small non-protein-coding RNAs. MiRNAs regulate the expression of target genes involved in the differentiation, proliferation, apoptosis, and drug resistance of BC and have the potential as biomarkers and therapeutic targets for BC [161]. MiRNAs can function as oncogenes by inhibiting tumor suppressor genes or miRNAs via oncogene silencing. MiRNA expression is commonly dysregulated in patients with many cancers, including BC. Many published studies have pointed to the role of miRNAs in BC progression, including in cell proliferation, differentiation, invasion, angiogenesis, metabolism, and drug resistance. The role of miRNAs in BC resistance to anti-estrogen therapy was only recently reviewed. Some of these miRNAs are directly linked with and regulated by E2 and/or indirectly regulate the expression of estrogen-sensitive genes [162,163].

The role of miRNAs in enhancing anti-estrogen therapy resistance is also associated with the regulation of ERα expression and function. MiRNAs, including miR-221/222, miR-342-3p, miR-873, and miR-22-3p, reduce ERα protein expression, and decreased ERα expression renders BC cells resistant to anti-estrogen therapy [153,164,165]. MiR-221/222 directly targets the 3′ unstranslated region of ERα mRNA and reduces protein expression. MiR-221 and miR-222 are overexpressed in tamoxifen-resistant BC cells [166] and play a role in the development of tamoxifen resistance through inhibition of the cell-cycle inhibitor p27/kip1. When miRNA-221/222 inhibits p27/kip1, binding of p27/kip1 to the CDK2/cyclin-E complex is inhibited, making progression through the cell cycle easier and promoting cell growth even if the ERs are occupied by tamoxifen [167]. Furthermore, miR-346 targets the transcriptional co-repressor receptor-interacting protein 140 in endocrine-resistant BC cell lines, suggesting a role for miRNAs in regulating estrogen-responsive genes [168]. Gonzalez et al. [169] identified miR-30a-3p, miR-30c, and miR182 as biomarkers of favorable prognosis for and clinical outcome of ER+ BC treated with tamoxifen. Moreover, overexpression of miR-205 is reported to mediate AI resistance in aromatase-overexpressing MCF-7 cells and to be associated with poor prognosis for breast tumors [170]. Researchers found that although the miRNA miR-342-3p did not directly target ERα, loss of miR-342-3p resulted in tamoxifen resistance of MCF-7 cells. However, overexpression of miR-342-3p sensitized MCF-7 cells to tamoxifen-mediated apoptosis [171].

### 6.10. Dysregulation of Long Noncoding RNAs (LncRNAs)

Emerging evidence implicates that long noncoding RNAs (lncRNAs) occupying the largest proportion of the noncoding transcriptome are involved in anti-estrogen resistance in ER+ BC through multiple mechanisms of action [172]. Recently, studies have suggested that some lncRNAs control many hallmarks of cancer progression, including cell proliferation, survival, metastasis, and drug resistance, and can serve as biomarkers or attractive therapeutic targets [173,174,175,176]. Therefore, an in-depth understanding of the contribution of lncRNAs in resistance to anti-estrogen therapies may improve the clinical outcomes of BC patients. In this section, we summarize lncRNAs studied in breast cancer cell models sensitive or resistant to anti-estrogen treatments and in clinical samples of breast tumors.

Of the multiple cancer-associated lncRNAs, HOTAIR (HOX antisense intergenic RNA) was among the most overexpressed in breast cancer. Studies have shown that HOTAIR induces invasion of breast carcinoma cells [177], and high HOTAIR expression levels associate with poor prognosis in breast cancer [178,179]. In addition, these studies reported that high HOTAIR expression is associated with ER and PR positivity, and HOTAIR expression is a strong predictor of poor clinical outcome, especially in ER+ breast cancer [178,179]. Xue et al. [174] reported that HOTAIR is overexpressed in tamoxifen-resistant ER+ breast cancer cells and tissues compared to its non-resistant counterparts and silencing of HOTAIR greatly reduces Tamoxifen-resistant MCF-7 cell proliferation. Moreover, this study revealed that HOTAIR interacts directly with the ER protein to increase ER transcriptional activity and as such ligand-independent breast cancer growth [174]. Zhang et al. [180] found that a novel anti-estrogen therapy-related ESR1/miR-130b-3p/HOTAIR network significantly correlated with poor survival in breast cancer patients with following systemic treatment (anti-estrogen therapy). Rongzhao et al. [181] showed that higher circulating HOTAIR levels correlated with larger tumor size, more positive lymph nodes, and more distant metastasis. Moreover, the survival analysis of breast cancer patients demonstrated that patients with high circulating HOTAIR levels had a poor disease-free survival than those with low circulating HOTAIR [181].

Thymopoietin antisense transcript 1 (TMPO-AS1) has been noted as another lncRNA reported to increase anti-estrogen resistance in breast cancer [175]. Expression of TMPO-AS1 is estrogen-inducible and increased in MCF-7 cells resistant to anti-estrogen therapy compared to levels in therapy-sensitive cells. Furthermore, it has been reported that TMPO-AS1 supports the proliferation of ER+ breast cancer cells in vitro and in vivo and is closely related to the estrogen signaling pathway [175].

The ADAM metallopeptidase with thrombospondin type 1 motif 9 (ADAMTS9) antisense RNA 2 (ADAMTS9-AS2) is lncRNA that functions as a tumor suppressor and reduces tamoxifen resistance [176]. ADAMTS9-AS2 has been shown to be expressed at low levels in BC tissues and tamoxifen-resistant MCF-7 cells, and the interaction between microRNA-130a-5p and ADAMTS9-AS2 is negatively correlated. Overexpression of ADAMTS9-AS2 reversed oncogenic roles induced by microRNA-130a-5p. In addition, PTEN inhibition reversed the decreased cell viability and induced apoptosis caused by ADAMTS9-AS2 overexpression. Taken together, low expression of ADAMTS9-AS2 enhances tamoxifen resistance through inhibition of PTEN and by targeting microRNA-130a-5p [176].

Urothelial carcinoma-associated 1 (UCA1) is a lncRNA known as oncogenic in breast cancer [182,183,184]. Studies conducted in ER+ tamoxifen-sensitive cells, such as MCF-7 and T47D, and tamoxifen-resistant cells derived from these cells, such as LCC2 and LCC9, showed that lncRNA UCA1 was upregulated in tamoxifen-resistant cells compared to tamoxifen-sensitive cells, and UCA1 confers tamoxifen resistance to breast cancer cells via the mTOR, Wnt/β-catenin, and PI3K/AKT signaling pathways [184,185,186].

Another lncRNA is breast cancer anti-estrogen resistance 4 (BCAR4), which has been identified as a predictor for tamoxifen resistance [187,188]. Studies in cell line and human breast tumors reported that BCAR4-mediated anti-estrogen therapy resistance is driven by HER2/ErbB2, ErbB3, and ErbB4 signaling [188,189,190].

Metastasis-associated lung adenocarcinoma transcript 1 (MALAT1) is a lncRNA that has been dysregulated in anti-estrogen therapy-resistant breast cancer [191]. MALAT1 has been found to act both oncogenic [192,193,194] and tumor suppressor [195] in breast cancer. High expression of MALAT1 is associated with short relapse-free survival in patients with ER+ breast cancer treated with tamoxifen. However, high MALAT1 expression is associated with poor relapse-free survival in patients with ER-negative breast cancer, implying that MALAT1 acts independently of the ER [191].

The large intergenic noncoding RNA-regulator of reprogramming (lincRNA-ROR) is another lncRNA that drives anti-estrogen therapy resistance. lincRNA-ROR increased tamoxifen resistance by suppressing autophagy in ER+ cells [196]. Moreover, lincRNA-ROR promotes estrogen-independent growth and tamoxifen resistance mediated by the MAPK/ERK signaling pathway in breast cancer [197].

The lincRNA termed lncRNA in non-homologous end joining (NHEJ) pathway 1 (LINP1), which was first determined to be overexpressed in triple-negative breast cancer, improves tamoxifen resistance in ER+ breast cancer [198]. LINP1, which promotes the proliferation of ER-positive MCF7 breast cancer cells, is negatively regulated by estrogen and upregulated in MCF7 and T47D and tamoxifen-resistant breast cancer cells derived from these cells. It has been suggested that LINP1 inhibits the ER signaling pathway by downregulating ERα as the mechanism that causes tamoxifen resistance [198,199].

The screening study identified cytoskeletal regulatory RNA (CYTOR) as the most significantly elevated lncRNA in established tamoxifen-resistant MCF7 cell lines compared with parental MCF7 cells. Consistently, higher CYTOR expression was detected in the tissues of tamoxifen-resistant patients compared to the tissues of patients not receiving tamoxifen therapy [200]. Silencing of CYTOR restored tamoxifen sensitivity of tamoxifen-resistant breast cancer cells. In addition, it has been noted that microRNA (miR)-125a-5p binds to CYTOR, and the expression of miR-125a-5p is negatively correlated with CYTOR in breast cancer tumor tissues [201]. Taken together, the data showed that lncRNAs have a crucial role as clinical markers in mediating tamoxifen resistance in ER+ BC.

## 7. Current and Potential Treatments to Improve Efficacy of Anti-Estrogen Treatments

### 7.1. SERDs

ERs remain a key driver and a therapeutic target for ER+ BC even after the development of resistance to tamoxifen. Therefore, because they reduce ER activity, SERDs represent an important class of ER antagonists for the treatment of metastatic ER+ BC in early-stage and tamoxifen-resistant cases. The SERD fulvestrant has been approved by the FDA for more than 10 years for the treatment of metastatic ER+ BC in postmenopausal women after tumor progression who have received early-stage AIs and tamoxifen [202]. Fulvestrant is an ERα antagonist that blocks ER signaling. It binds to the ER with greater affinity than does tamoxifen, preventing ER dimerization and translocation to the nucleus and initiating transcription of target genes [203,204]. However, the ER-fulvestrant complex is reversible and allows degradation of the ER protein by directing it to the ubiquitin-proteasome system [205,206]. Fulvestrant exhibits bioactivity at high doses of about 500 mg per month and limits its therapeutic bioavailability due to the high dose. This disadvantage of fulvestrant has prompted researchers to explore new oral SERDs [202]. New SERDs that can also degrade ERα, reduce target protein expression, and block ER signaling are currently under development. Structurally different from fulvestrant, these novel SERDs can be classified into two groups according to the chemical structures of their side chains. Specifically, GDC-0810, AZD9496, LSZ102, and G1T48 (rintodestrant) are characterized as nonsteroidal acrylic acid side chain SERD compounds. RAD1901 (elacestrant), GDC-0927, GDC-9545, SAR439859 (amcenestrant), and AZD9833 (camizestrant) are classified as nonsteroidal analogs with basic amino side chains [207,208]. Novel SERDs have entered clinical trials, including LSZ102, GDC-0810, AZD9496, AZD9833, rintodestrant, GDC-0927, GDC-9545, elacestrant, and SAR439859 [208,209,210,211].

The phase I/Ib clinical trial showed that LSZ102 plus the ribociclib produced a better median progression-free survival duration than did LSZ102 alone in patients with advanced ER+ BC who progressed after anti-estrogen therapy (6.2 (95% CI, 5.6–6.4) vs. 1.8 months, (95% CI, 1.7–2.5)). Moreover, LSZ102 in combination with alpelisib produced a better median pro-gression-free survival duration than did LSZ102 alone (3.5 (95% CI, 3.2–5.5) vs. 1.8 months (95% CI, 1.7–2.5)) [212]. In the first report, LSZ102 plus ribociclib or alpelisib showed encouraging clinical activity in ER+ BC patients, but the clinical trial was recently terminated (NCT02734615).

Recently, phase I clinical trial results of RAD1901 have been published in pretreated (SERD or CDK4/6 inhibitor) ER+ metastatic breast cancer patients [213]. The objective response rate (ORR) in patients receiving 400 mg once daily was 15.0% for patients with prior SERD, 16.7% for patients with prior CDK4/6 inhibitor, and 33.3% for patients with an ESR1 mutation. The clinical benefit rate was 42.6% and related to a decrease in ESR1 mutation. Phase III clinical trial of RAD1901 is ongoing. This is significant as it is the first oral SERD tested in a phase III study (NCT03778931).

GDC-9545 is currently being evaluated for efficacy and safety in multiple clinical trials: GO39932 is a phase Ib/II clinical trial evaluating the efficacy of GDC-9545 alone and in combination with palbociclib or an LHRH agonist in patients with metastatic ER+ breast cancer (NCT03332797). Patients from GDC-9545 monotherapy exhibited a median ORR of 13% (95% CI, 4–30%) and a median PFS of 7.8 months (95% CI, 5.3–11.4). In patients treated with GDC-9545 plus palbociclib, the median ORR was 33% (95% CI, 20-49%) and the median PFS was 9.3 months (95% CI, 8.9% not evaluable). GDC-9545 was well tolerated in ER+ metastatic breast cancer as a single agent and, in combination with palbociclib, promoted pharmacokinetic, pharmacodynamic, and anti-tumor activity and is recommended for phase III clinical trials [200]. AcelERA phase II trial is evaluating the efficacy and safety of GDC-9545 compared with the physician’s choice of endocrine monotherapy in patients with ER+ locally advanced or metastatic breast cancer treated one or two prior lines of systemic therapy in the locally advanced or metastatic setting (NCT04576455). The clinical trial of CoopERA phase II evaluated the efficacy, safety, and pharmacokinetics of GDC-9545 plus palbociclib compared with anastrozole plus palbociclib for postmenopausal women with ER+ untreated early BC, and results have not yet been posted (NCT04436744). Another phase III clinical trial (persevERA) is currently being evaluated for efficacy and safety of GDC-9545 combined with palbociclib compared with letrozole combined with palbociclib in patients with ER+, locally advanced or metastatic BC (NCT04546009). The preoperative WOO study of GDC-9545 in untreated, operable ER+ BC was completed. Results were not fully released, but based on data from the 2021 ASCO meeting, GDC-9545 was well tolerated in the preoperative setting in ER+ operable BC, and pharmacodynamics were consistent with the 30mg dose providing maximum ER inhibition (NCT03916744) [214].

SAR439859 has entered several clinical trials in patients with ER+ breast cancer as monotherapy and combination therapy. The ongoing phase I/II trial of AMEERA-1 is testing once-daily SAR439859 in pretreated (including targeted therapies and fulvestrant) postmenopausal patients with ER+ advanced BC (NCT03284957). In the Part A dose escalation phase, patients were treated with SAR439859 20–600 mg once daily. Based on the absence of dose-limiting toxicities and pharmacokinetics, 400 mg once daily was selected as the Phase 2 dose for the Part B dose expansion phase. The study results confirmed objective response rate (ORR) was 10.9%; the overall clinical benefit rate was 28.3%. The CBRs of patients with wild-type and mutated ESR1 were 9/26 (34.6%) and 4/19 (21.1%), respectively. In conclusion, SAR439859 400 mg once daily for monotherapy in phase II is well tolerated without dose-limiting toxicities and shows pro-antitumor activity regardless of baseline ESR1 mutational status [215].

Phase II Study of AMEERA-3 is ongoing in locally advanced or metastatic ER+ breast cancer patients and testing SAR439859 in comparison to endocrine monotherapy of the choice of the physician (NCT04059484). Phase II window study of AMEERA-4 tests SAR439859 versus letrozole in post-menopausal patients with ER+ pre-operative BC. This clinical trial was terminated early based on a strategic sponsor decision, not due to any safety concerns (NCT04191382). Lastly, the AMEERA-5 study is assessing SAR439859 in combination with palbociclib versus letrozole plus palbociclib as a first-line therapy for patients with ER+ metastatic BC (NCT04478266).

The clinical trial of G1T48 alone or in combination with palbociclib in endocrine-resistant metastatic patients is ongoing, and results have not yet been released (NCT03455270).

SERENA-1, which is a multicentre dose escalation and expansion study, is currently testing AZD9833 alone or in combination with palbociclib or everolimus or abemaciclib capivasertib in women with endocrine-resistant ER+ breast cancer (NCT03616587). Preliminary results showed that ORR and CBR for the monotherapy group were 10.0% and 35.3%, respectively, and the median PFS was 5.4 months. In patients treated with AZD9833 plus palbociclib combination therapy, the ORR and CBR were 14.3% and 71.4%, respectively. The tolerability of AZD9833 plus palbociclib was consistent with the observed tolerability profile of AZD9833 monotherapy and the known tolerability profile of palbociclib. Pharmacokinetic analysis showed similar AZD9833 exposure for monotherapy and palbociclib combination therapy. No patient required discontinuation of camizestrant treatment due to an adverse event associated with AZD9833 [216,217]. Based on promising early clinical data with AZD9833; a phase 2 study comparing the efficacy and safety of AZD9833 with fulvestrant (SERENA-2; NCT04214288), a pre-surgical “window of opportunity” study (SERENA-3; NCT04588298), and a phase III study comparing the effects of AZD9833-palbociclib combination with the anastrozole–palbociclib combination (SERENA-4; NCT04711252) is ongoing in women with ER+ advanced BC.

### 7.2. Targeting PI3K/AKT/mTOR Signaling

Around 50% of breast tumors have hyperactivation of PI3K/AKT/mTOR signaling, which promotes cancer cell survival, proliferation, cell metabolism, drug resistance, and angiogenesis. PI3K/AKT/mTOR signaling is activated in advanced ER+ BC; thus, this signaling is proposed to be an important target to overcome ER+ BC resistance [218,219]. In addition, preclinical and clinical evidence demonstrates that inhibition of PI3K/AKT/mTOR signaling may increase the efficacy of anti-estrogen therapy for ER+ BC [220]. PI3K/Akt/mTOR signaling interacts directly or indirectly with estrogen and/or ERs. First, it activates estrogen-independent ER transcriptional activity. In addition, activation of the estrogen pathway triggers the synthesis of many components of the PI3K/AKT/mTOR pathway [218]. Moreover, preclinical studies have shown that AKT can activate ER signaling independently of estrogen and mTOR inhibitors combined with anti-estrogen therapy can overcome hormone therapy resistance of BC [221]. Mutations or amplifications of the genes encoding the catalytic subunits p110a (PIK3CA) and p110β (PIK3CB), as well as the regulatory subunit p85a (PIK3R1) in the PI3K pathway, lead to many changes in BC [222]. PIK3CA mutations that can trigger the α isoform of PI3K (p110α) have been demonstrated in about two out of every five patients with BC. In addition, the FDA has approved testing of BC patients with PIK3CA mutations using DNA isolated from breast tumors and/or plasma specimens [223,224].

When combined with inhibitors of the PI3K/AKT/mTOR pathway, the efficacy of anti-estrogen therapies enhanced in anti-estrogen therapy-resistant BC patients. Researchers clinically evaluated the mTOR inhibitor everolimus and PIK3K inhibitors (e.g., alpelisib), which selectively target p110α (encoded by PIK3CA) [225]. Everolimus, an analog of rapamycin, has been approved by the FDA as an oral antitumor agent for ER+/HER- BC [226]. Everolimus acts on mTOR by binding to the cyclophilin FKBP-12, thereby acting as an inhibitor of downstream signaling [227,228]. Consisting of two complexes (mTORC1 and mTORC2), mTOR is the downstream target of PI3K/AKT. Whereas mTORC1 increases cell growth by stimulating mRNA translocation, protein synthesis, and lipid synthesis, mTORC2 stimulates AKT phosphorylation. Rapalogs target mTORC1 similarly to rapamycin and can phosphorylate the ERs via ribosomal S6 kinase 1, meaning that Rapalogs lead to ligand-independent ER activation [227].

Everolimus is superior to rapamycin in terms of pharmaceutical properties [227]. However, the effect of rapalogs on mTORC2 is very controversial. Whereas some think that everolimus may exert its effects on both mTOR complexes, most experts suggest that it only acts on mTORC1 [229]. Bachelot et al. [230] showed that in postmenopausal women with AI-resistant metastatic BC, treatment with everolimus combined with tamoxifen produced better clinical benefits, time to progression, and overall survival than did treatment with tamoxifen alone. In preclinical studies, the combined use of everolimus and AIs had a synergistic effect, resulting in the inhibition of proliferation and increased apoptosis [231]. In a phase 2 study comparing neoadjuvant everolimus plus letrozole with letrozole alone in patients with newly diagnosed ER+ BC, the response rate was higher for the combination than for the monotherapy (NCT00107016) [232]. Baselga et al. [228] evaluated the effects of everolimus and the AI exemestane in 724 postmenopausal women with advanced hormone receptor-positive BC (NCT00863655). The combination of everolimus and exemestane was found to have a higher PFS than exemestane monotherapy. Due to the highly successful PFS rates, exemestane and everolimus combination therapy is approved for patients with advanced ER+ BC in the USA and Europe.

GINECO phase II trial assessed the efficacy and safety of everolimus in combination with tamoxifen in patients with hormone receptor-positive metastatic BC resistant to AIs (NCT01298713). Patients who received combination treatment exhibited an improved clinical benefit rate, time to progression, and overall survival compared to patients who received tamoxifen alone [230].

The clinical trial of BELLE-2 tested the pan-PI3K inhibitor buparlisib plus fulvestrant versus fulvestrant plus placebo in postmenopausal women with hormone receptor-positive, locally advanced or metastatic BC refractory to aromatase inhibitör (NCT01610284). PFS improved in patients receiving combination therapy compared to monotherapy, but the high toxicity warrants the use of other more α specific PI3K inhibitors [233].

The clinical trial of SOLAR-1 evaluated the efficacy of alpelisib (a PI3Kα specific inhibitor) plus fulvestrant in men and postmenopausal women with advanced BC progressing on or after prior aromatase inhibitor therapy (NCT02437318). A significant improvement in PFS has been reported in patients with PIK3CA mutations receiving alpelisib therapy [234].

In summary, investigation of mutations encountered in the PI3K/AKT/mTOR signaling pathway and inhibitors of this pathway may provide a real benefit in hormone receptor-positive BC survival. Therefore, future efforts should be directed toward reducing the negative side effects of dual combination therapies to increase the efficacy of these inhibitors. Moreover, different combinations of these signaling inhibitors are promising in therapy for BC resistant to anti-estrogen therapy.

### 7.3. Targeting CDK4/CDK6

Dysregulation of cell-cycle control is commonly detected in many cancers and promotes cancer cell proliferation and malignant transformation [235]. Therefore, targeting cell-cycle checkpoint regulators can be an important therapeutic strategy for BC alone and in combination with anti-estrogen therapies and can increase the efficacy of anti-estrogen therapies. Selectively interrupting cell-cycle regulation in cancer cells by interfering with serine/threonine CDKs results in the death of cancer cells without toxic effects on normal cells [236]. Activation of growth factors and ERs activates the CDK4/CDK6/cyclin D complex via the PI3K/AKT/mTOR, MAPK/Ras/Raf, and Wnt/β-catenin pathways [237]. This activation regulates pRb, which is critical for the transition from the G1 phase to the S phase. When pRb is activated, it blocks the E2F family of transcription factors and arrests the G1-to-S transition, which inhibits cell-cycle progression. pRb is phosphorylated and inactivated by the CDK4/CDK6/cyclin D complex, and E2F transcription factors are released, enabling S-phase entry [238]. The CDK4/CDK6/cyclin D complex is negatively regulated by the INK4 and Cip/Kip protein families, which are endogenous CDK inhibitors [239].

Abnormal activation of the CDK4/CDK6/cyclin D/pRb pathway, particularly the cyclin D/CDK4 complex, is seen in about 70% of BCs [240]. Preclinical studies showed that hyperactivity of the CDK4/CDK6/cyclin D complex leads to uncontrolled cell proliferation, and pharmacological inhibition of this triad is an effective and promising therapeutic strategy for ER+ BC. In addition, CCND1 (the gene encoding cyclin D1), which is overexpressed in ER+ BC patients, is a direct target of the ERs, and activation of the CDK4/CDK6/cyclin D complex contributes to anti-estrogen therapy resistance of ER+ BC [241]. The combination of CDK4/CDK6 inhibitors and known anti-estrogen agents is synergistic in the preclinical setting, and this has been confirmed in large randomized clinical trials [242,243,244]. The combination of standard anti-estrogen therapy with these inhibitors improved survival rates in patients with ER+ BC [242,245,246].

Recently, the clinical success of CDK4/6 inhibitors (e.g., palbociclib, ribociclib, abemaciclib) in combination with AIs (e.g., letrozole) or SERDs (e.g., fulvestrant) in the treatment of advanced hormone-resistant and metastatic ER+ BC led to approval by the FDA of CDK4/6 inhibitors [247]. The PALOMA-3 trial showed that fulvestrant plus the CDK4/6 inhibitor palbociclib produced a better median progression-free survival duration than did fulvestrant alone (9.2 vs. 3.8 months; hazard ratio [HR], 0.42 [95% CI, 0.32–0.56]; *p* < 0.001) [248]. Moreover, in patients with metastatic ER+ BC that had progressed during anti-estrogen therapy, abemaciclib in combination with fulvestrant produced a better median progression-free survival duration than did fulvestrant alone (16.4 vs. 9.3 months; HR, 0.55 [95% CI, 0.45–0.68]; *p* < 0.001) [249]. In the MONALEESA-7 clinical trial, ribociclib in combination with anti-estrogen therapy resulted in a better overall survival rate at 42 months than did anti-estrogen therapy alone (tamoxifen, letrozole/AI, or goseralin [Zolodex]; 70.2% vs. 46.0%; HR for death, 0.71 [95% CI, 0.54–0.95]; *p* < 0.01) [250].

In the MONARCH clinical trial, the median overall survival duration in the abemaciclib group was 46.7 months compared with 37.3 months in the placebo group (HR, 0.757 [95% CI, 0.606–0.945]; *p* = 0.01) [251]. When used as first-line therapy for metastatic ER+ BC, abemaciclib combined with an AI also resulted in better progression-free survival than did an AI alone [252]. Given the findings of these trials, adding CDK4/6 inhibitors to anti-estrogen therapy is now the standard of care for patients with advanced metastatic ER+ BC. The most common adverse events of the CDK4/6 inhibitors in these trials were neutropenia (20–60%) and leukopenia (7–21%) with ribociclib, diarrhea (10–80%) and transaminitis (30–50%) with abemaciclib. Notably, ribociclib can prolong the QTc interval. Biomarkers that are clinically useful in determining which BC patients will have good responses to CDK4/6 inhibitors have yet to be identified.

### 7.4. Combination of CDK4/6 and PI3K/AKT/mTOR Inhibitors

The results observed with CDK4/6 inhibitors have been considered promising and approved by the FDA, but relapses continue to occur in a large number of ER+ BC patients [253,254]. Clinical outcomes of monotherapy PI3K inhibitors have been modest to date [255]. A combinatorial drug screen on PIK3CA mutant cancers resistant to PI3K inhibitors revealed that combined inhibition of CDK 4/6 and PI3K synergistically reduced cell viability and improved tumor regressions [256]. Studies have suggested that inhibition of CDK 4/6 alleviates acquired and intrinsic resistance to PI3K inhibitors and delays anti-estrogen resistance [257]. Combination inhibition of PI3K and CDK4/6 overcomes the resistance resulting from monotherapy inhibition of CDK4/6 by downregulating cyclin D1 and arresting cell cycle progression [258].

Currently, several clinical trials are enrolled in triplet combination of CDK 4/6 inhibitors, anti-estrogen therapy and PI3K/AKT-mTOR inhibitors (NCT02871791; NCT02684032; NCT02732119; NCT01857193; NCT02057133). The clinical trial of BYLIEVE phase II, including ER+ advanced breast cancer patients whose disease has progressed on or after prior CDK4/6 inhibitor (CDK4/6i) combination therapy, is an ongoing multicenter study assessing the combination therapy of alpelisib and endocrine therapy (either fulvestrant or letrozole) [259]. Preliminary results exhibited clinically meaningful efficacy of alpelisib plus anti-estrogen therapy after prior treatment with CDK4/6 inhibitors [259]. A phase I clinical trial evaluates the efficacy of gedatolisib, a dual inhibitor of PI3K/mTOR, in combination with palbociclib/letrozole or palbociclib/fulvestrant in patients with metastatic breast cancer. Although the results of the clinical trial have not yet been fully posted, the clinically relevant anti-tumor activity and safety profile of this combination therapy was presented at the 2018 ASCO meeting [260]. The clinical trial of TRINITI-1, including ER+ advanced breast cancer patients, is a Phase I/II trial of ribociclib with everolimus + exemestane. TRINITI-1 met the primary efficacy endpoint and demonstrated the clinical benefit and tolerability of triple therapy with ribociclib plus everolimus plus exemestane in patients with ER+ BC resistant to anti-estrogen therapy [261].

Ongoing/completed clinical trials of anti-estrogen therapy-resistant BC is summarized in Table 1.

### 7.5. Novel RNA Interference-Based Therapies That Reverse Anti-Estrogen Therapy Resistance of ER+Breast Cancer

Silencing specific tumorigenic genes via RNA interference is rapidly evolving as a targeted approach to cancer therapy. Small interfering RNAs (siRNAs), miRNAs, and long noncoding RNAs (lncRNAs), which are effector molecules that act on target genes, can be used to downregulate or “turn off” specific cancer genes [262]. For maximum benefit in RNA interference-based therapy, genes belonging to different signaling pathways that support the proliferation of cancer cells have been targeted using specific siRNAs [263]. The targeted aim of RNA interference is to downregulate or silence the expression of genes important for cancer cell survival, induce specific destruction of pathological cells, and overcome multidrug resistance of cancer cells to anti-estrogen therapy, chemotherapy, and radiotherapy [264,265].

#### 7.5.1. Therapeutic Targets That Potentially Reverse Anti-Estrogen Therapy Resistance

To identify potential therapeutic targets in BC patients with anti-estrogen therapy resistance, we extensively searched the literature and identified the target genes and proteins whose inhibition by siRNA-based therapeutics can overcome anti-estrogen therapy resistance (Table 2).

SiRNA-induced targeting ERα has proven to be effective in ER+ BC cells; silencing of ERα suppresses BC cell proliferation by inducing cell-cycle arrest and cell [282]. Zhang et al. [266] showed that the knockdown of MED1, the transcriptional co-activator of ERα, via siRNA sensitized ERα+ BC cells to treatment with tamoxifen and fulvestrant, and clinical data revealed that MED1 expression is associated with endocrine therapy resistance of BC. The depletion of TRIM56 with siRNA, which is associated with the AF-1 domain of ERα and is a regulatory factor in ERα signaling, reduced the proliferation of ER+ BC cells in vitro and in vivo. Authors have reported that TRIM56 is associated with poor prognosis for ER+ BC [267]. In an in vitro siRNA screen, the knockdown of phosphoinositide-dependent kinase 1 (PDK1), which is involved in PI3K/AKT signaling, increased the sensitivity of BC cells to CDK4/6 inhibitors and synergistically inhibited cell proliferation [274]. SiRNA kinome-based knockdown in MCF-7 and T47D cells of Wee1, a cell-cycle regulator that functions at the G2/M checkpoint, helped reverse resistance of cells to the CDK4/6 inhibitor palbociclib. In the same study, the depletion of CDK7 amplified the effect of palbociclib on ER+ BC cells [268]. In a study in which researchers targeted MST2, which mediates BC progression, using siRNA, the proliferation of MCF-7 cells was significantly suppressed. Moreover, MST2 knockdown promoted caspase-mediated apoptosis of MCF-7 cells and delayed tumor growth [269]. In vitro, effective siRNA-mediated reduction of ERα protein expression in MCF-7 cells resulted in an increase in the antiproliferative activity. [270]. Inhibition of CDK2 by siRNA has reversed the palbociclib resistance of MCF-7 cells [271]. CDK6 inhibition by siRNA increased the sensitivity of ER+ cancer cells to ribociclib [283]. The PI3K pathway kinase PDK1 was highly upregulated in ribociclib-resistant ER+ cells, and siRNA-mediated downregulation of PDK1 sensitized the cells to ribociclib [274]. Furthermore, siRNA silencing of the zinc finger protein ZNF213, which is associated with poor prognosis in endocrine-treated BC patients and regulates the stability of ERα, inhibited the expression of ERα target genes in MCF7 cells and proliferation of MCF7 and T47D cells [276]. Knockdown of the nuclear envelope protein LEM4, which binds to ERα and regulates its transactivation, and combination therapy with tamoxifen and a CDK4/6 inhibitor overcame tamoxifen resistance of MCF7-TAMR cells [273].

#### 7.5.2. MicroRNA-Based Therapeutics That Potentially Reverse Anti-Estrogen Therapy Resistance

With the incorporation of miRNAs into the molecular mechanisms that contribute to tamoxifen resistance, miRNAs have received much attention as new therapeutic targets for BC [284,285]. miRNA dysregulation in BC was first reported in 2005, and since then, more than 1000 studies have focused on miRNAs [286]. Some of these miRNAs are E2-sensitive and -regulated and affect the expression of estrogen target genes [163]. In particular, oncomiRs have begun to attract great interest as treatment-improving targets [287]. Various synthetic molecular inhibitors, such as antagomiRs and miRNA sponges, have shown promise in inhibiting and antagonizing the cellular function of miRNA. Unlike antagomiRs, which inhibit only single, specific miRNAs, the synthetic miRNA sponges contain multiple antisense sites that can bind to multiple miRNAs [288]. Developing miRNA sponge technology to silence oncomiRs that contribute to tamoxifen resistance is a logical therapeutic strategy to restore tamoxifen sensitization of BC and elucidate the functions of these miRNAs [289].

Most studies of miRNAs related to tamoxifen resistance have focused on the oncomiR family members miR221/222. The target of miR221/222 is the ESR1 gene, which reduces ERα translation, and tamoxifen-resistant cells have high miR-221/222 expression [166,290,291]. Numerous miRNAs associated with oncogenic signaling other than miR221/222 have been identified. Kim et al. [292] reported that miR-192-5p caused tamoxifen resistance by decreasing ERα expression. They also showed that miR-500a-3p increased sensitivity to tamoxifen by acting in a nontumorigenic manner. In another study, researchers noted that miR-335 transcription plays an oncogenic role by triggering resistance of estrogen-responsive BC to anti-estrogen therapy [293]. Ikeda et al. [285] found that miR-378a-3p was downregulated in both tamoxifen-resistant BC cells and breast tumor specimens. In addition, they reported that low miR-378a-3p expression was associated with tamoxifen resistance and poor prognosis. Wei et al. [294] showed that miR-484 is expressed at low levels in tamoxifen-resistant MCF-7 cells and cancer stem cells and that overexpression of miR-484 enables cancer stem cells to restore tamoxifen sensitivity.

Another important miRNA acting as a mediator of anti-estrogen therapy resistance is miR-22. Investigators found that suppression of miR-22 expression with a synthetic miR-22 inhibitor remarkably repressed fulvestrant-resistant 182^R^-6 BC cell proliferation, promoted apoptosis, and caused S-phase arrest through upregulation of p21^Cip1/Waf1^ and p27^Kip1^ [295]. In another study using an miRNA array with fulvestrant-resistant BC cells, miR-375 expression was markedly lower in fulvestrant-resistant BC cells than in MCF-7 cells. Additionally, dual treatment with small-molecule EGFR-targeted inhibitors upregulated miR-375 in fulvestrant-resistant ER+ BC cells [296]. In a study by Yu and colleagues, UCP2 was identified as a direct target of miR-214. In addition, miR-214 sensitized ER+ BC cells to tamoxifen and fulvestrant through suppression of autophagy by targeting UCP2, a mitochondrial protein involved in the production of intracellular reactive oxygen species. Remarkably, they established a negative correlation between miR-214 and UCP2 mRNA expression in clinical BC tissue specimens [297]. In another study pointing out the effect of miRNAs on anti-estrogen therapy, MCF-7 cells with silenced miR-21 became more sensitive to tamoxifen and fulvestrant, and anti-estrogen treatment reduced miR-21 levels. Moreover, suppression of miR-21 triggered autophagic cell death [298]. These results highlight that targeting miRNAs that affect autophagy is promising for overcoming the resistance of BC to anti-estrogen therapy.

Yao et al. [299] reported that miR-27a-3p is upregulated in MCF-7 cells and that miR-27a-3p targets programmed death-ligand 1 and contributes to the escape of MCF-7 cells from the immune system via PTEN/AKT/PI3K signaling. Moreover, Liu et al. [300] reported that miR-575 expression is increased in ER+ BC cells and regulated by estrogen/ERα. In the same study, miR-575 expression decreased with anti-estrogen therapy, and p27, the key regulator of G1- to S-phase transition, antagonized ERα by targeting miR-575. Tokumaru et al. [301] showed that miR-143 suppressed MCF-7 cell growth by targeting KRAS, whereas miR-143 improved overall survival in ER+ BC patients. Moreover, they reported that increased expression of miR-143 had anticancer activity by triggering the infiltration of anticancer immune cells in the ER+ breast tumor microenvironment. Nair and colleagues studied miR-18a expression levels in MCF-7 and ZR-75-1 cells and ER+ human breast tumor specimens [302]. Overexpression of miR-18a-5p induced by a synthetic mimic in ER+ MCF-7 and ZR-75-1 cells reduced ER protein abundance.

Using clinical ER+ breast tumor specimens, Liu and colleagues found that miR-26a expression was lower in the tumor specimens than in matched normal breast tissue specimens. Additionally, they demonstrated that miR-26a targets the transcription factor E2F7 and decreases E2F7 protein expression in MCF-7 and T47D cells. Notably, suppression of E2F7 expression decreased MYC proto-oncogene expression in MCF-7 and T47D cells [303].

We conducted a search of the literature using the keywords “microRNA/miRNA” and “tamoxifen resistance/tam-resistance/endocrine resistance/anti-estrogen resistance.” Through this search, we obtained a list of miRNAs possibly associated with the anti-estrogen therapy resistance of BC. Information about these miRNAs is summarized in Table 3.

## 8. Conclusions

Resistance to anti-estrogen therapies for ER+ BC is common in the majority of patients and represents a major clinical problem. Therefore, the discovery of novel targets and the identification of new strategies to overcome this resistance is of great importance and requires a deeper understanding of the resistance mechanisms. Combination therapies targeting signaling pathways involved in resistance to hormonal therapies that synergistically block cancer cell survival pathways is an encouraging option going forward, especially for metastatic relapses. The use of agents targeting these pathways has produced promising results in preclinical models, but few of them have demonstrated efficacy in the clinical setting. In recent years, candidate cellular pathways such as CDK4/CDK6, IGF-IR, PI3K/AKT/mTOR, and MAPK have played a role in the development of anti-estrogen therapy resistance of BC and are proven targets for the development of new therapeutic strategies. The use of agents targeting these pathways also has produced promising results in preclinical models, but few have demonstrated efficacy in the clinical setting. More comprehensive studies are needed to identify marker genes critical for BC anti-estrogen therapy resistance development and to better understand the genetic background.

## Figures and Tables

**Figure 1 cancers-14-05206-f001:**
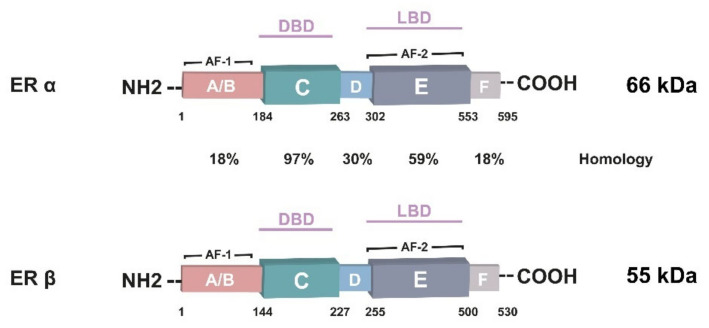
Schematic structure and sequence organization of ERα and ERβ. They differ in terms of amino acid numbers. ERα consists of 595 amino acids, while ERβ contains 530 amino acids. ERα and ERβ share identical characteristics. They are formed by different domains. Amino-terminal A/B domains share 18% amino-acid homology between the ERs. C region (97%) consists of DNA-binding domain. D domain (30%) contains a nuclear localization signal. Multifunctional carboxyl-terminal (E) domain, which shows 59% amino-acid identity between the ERs. The carboxyl-terminal F domain shares an 18% amino-acid identity. (DBD: DNA-binding domain; LBD: ligand-binding domain; AF-1: activation domain).

**Figure 2 cancers-14-05206-f002:**
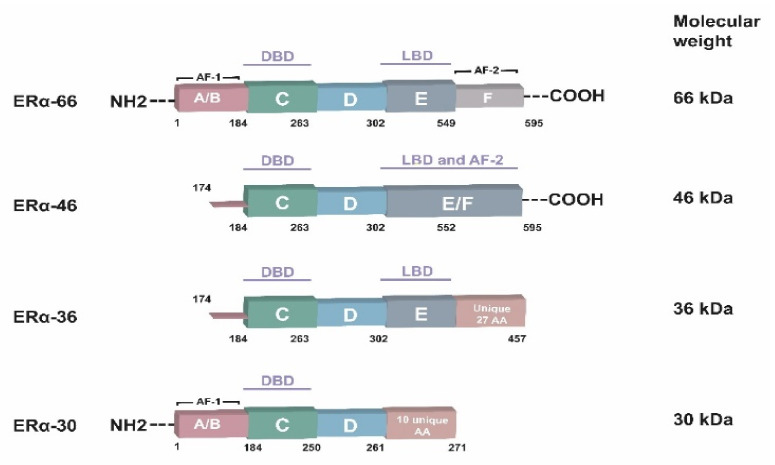
Schematic representation of ERα isoforms including ERα-66, ERα-46, ERα36, and ERα-30. (DBD: DNA-binding domain; LBD: ligand-binding domain; AF-1: activation domain-1; AF-2 activation domain-2).

**Figure 3 cancers-14-05206-f003:**
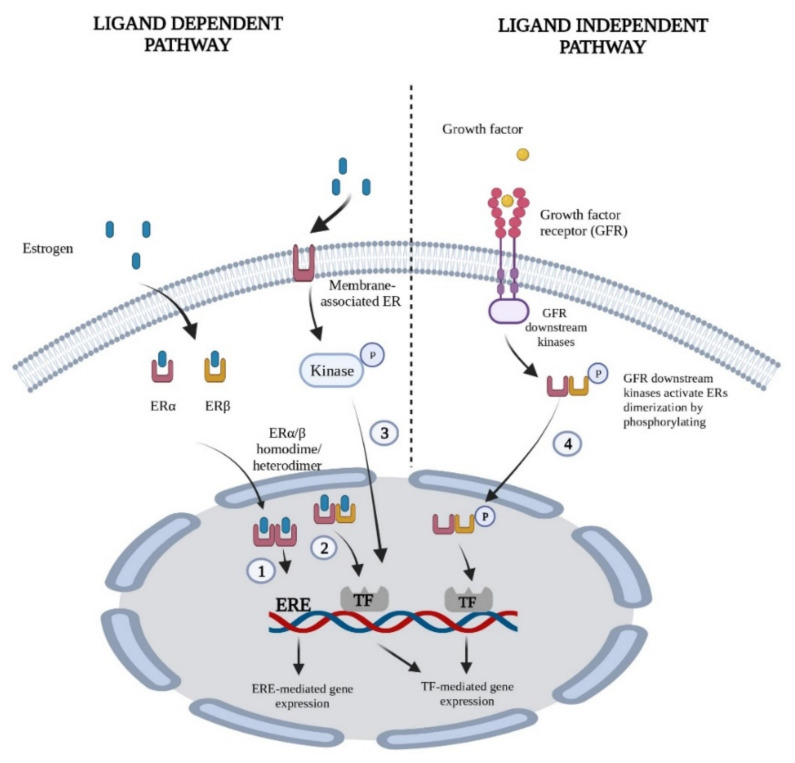
Illustration of ER signaling pathways. In ligand-dependent signaling, 1: Estrogen binds to intracellular ER and translocates to nucleus for binding to estrogen response elements (EREs) in target gene promoters regulating estrogen-responsive genes, 2: Estrogen binds to intracellular ER and translocates to nucleus for binding to transcription factors (TFs), 3: Estrogen binds to membrane-associated ER and induces signaling molecules, including Src tyrosine kinase, mitogen-activated protein kinase (MAPK), PI3K/AKT, and protein kinase C, regulating transcription of target genes by phosphorylating transcription factors. In ligand-independent signaling, 4: Growth factors bind to growth factor receptors (GFR) to activate intracellular kinases. GFR downstream kinases activate ERs dimerization by phosphorylation and regulate target gene expression in absence of estrogen in both ERE-dependent nuclear signaling or by interacting with other transcription factors.

**Figure 4 cancers-14-05206-f004:**
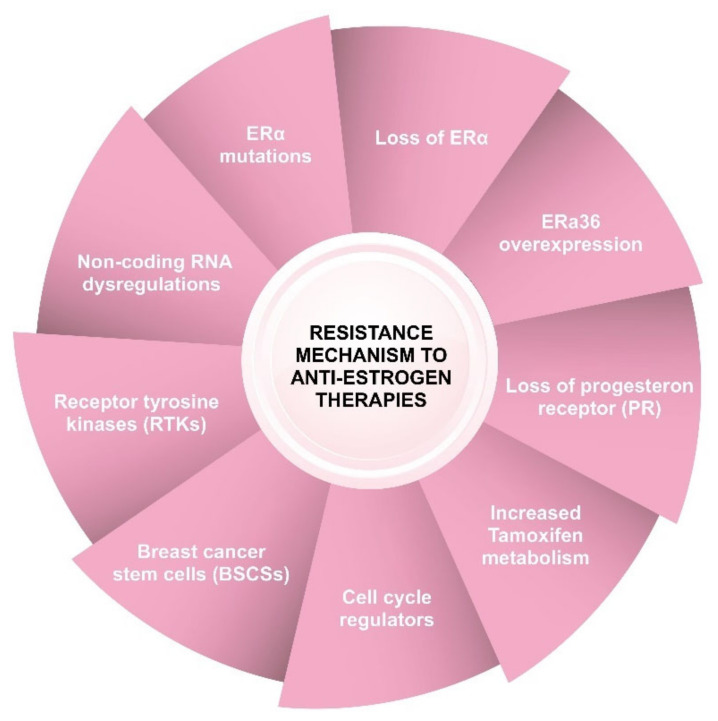
Resistance mechanisms to anti-estrogen therapies.

**Table 1 cancers-14-05206-t001:** Clinical trials of anti-estrogen therapy-resistant BC.

Drug	Treatments	Clinical Trial Phase	Clinical Trial Registry Number
SERDs			
Fulvestrant	Single agent	Phase III, active	NCT00099437
LSZ102	Single agent and combination with ribociclib or alpelisib	Phase I, terminated	NCT02734615
RAD1901 (Elacestrant)	Single agent	Phase III, active	NCT03778931
GDC-9545 (Giredestrant)	Single agent and combination with palbociclib and/or luteinizing hormone-releasing hormone (LHRH) agonist	Phase I, active	NCT03332797
	GDC-954 single agent compared with physician’s choice of endocrine monotherapy	Phase II, active	NCT04576455
	GDC-9545 plus palbociclib compared with anastrozole plus Palbociclib	Phase II, completed	NCT04436744
	GDC-9545 combined with palbociclib compared with letrozole combined with palbociclib	Phase III, recruiting	NCT04546009
	Monotherapy	Phase I, completed	NCT03916744
SAR439859 (Amcenestrant)	Single agent and in combination with other anti-cancer therapies	Phase I/II, active	NCT03284957
	SAR439859 in comparison to endocrine monotherapy of the choice of the physician	Phase II, active	NCT04059484
G1T48	Single agent and in combination with palbociclib	Phase I, active	NCT03455270
AZD9833	Single agent and combination with palbociclib or everolimus or abemaciclib or capivasertib	Phase I, recruiting	NCT03616587
	AZD9833 versus intramuscular (IM) fulvestrant	Phase II, active	NCT04214288
	Single agent	Phase II, recruiting	NCT04588298
	AZD9833 plus palbociclib versus anastrozole plus palbociclib	Phase III, recruiting	NCT04711252
PI3K/AKT/mTOR Inhibitors			
Everolimus (RAD001)	Single agent	Phase III, active	NCT01805271
	Everolimus in combination with letrozole	Phase II, completed	NCT01231659
	Everolimus combination with exemestane	Phase IV, completed	NCT01743560
	Everolimus plus letrozole versus neoadjuvant fluorouracil, epirubicin plus cyclophosphamide (FEC)	Phase II, completed	NCT02742051
	Everolimus combined with anti-estrogen therapy	Phase II, completed	NCT02291913
	Everolimus in combination with exemestane	Phase III, completed	NCT00863655
	Tamoxifen plus everolimus versus tamoxifen alone	Phase II, completed	NCT01298713
Buparlisib (BKM120)	Buparlisib plus fulvestrant versus fulvestrant plus placebo	Phase III, completed	NCT01610284
Alpelisib	Alpelisib plus fulvestrant	Phase III, active	NCT02437318
CDK4/CDK6 Inhibitors			
Palbociclib	Palbociclib in combination with fulvestrant	Phase III, active	NCT01942135
	Palbociclib with everolimus plus exemestane	Phase I/II, completed	NCT02871791
Ribociclib (LEE011)	Ribociclib in combination with anti-estrogen therapy	Phase III, active	NCT02278120
	Ribociclib with everolimus plus exemestane	Phase I/II, completed	NCT02732119
	Ribociclib with everolimus and exemestane	Phase I, completed	NCT01857193
	Triplet combination of ribociclib, everolimus and exemastane	Phase I/II, completed	NCT02732119
Abemaciclib (LY2835219)	Abemaciclib plus fulvestrant or fulvestrant alone	Phase III, active	NCT02107703
	Abemaciclib in combination therapies	Phase I, active	NCT02057133

**Table 2 cancers-14-05206-t002:** SiRNAs linked with anti-estrogen therapy resistance of ER+ BC.

Cancer Hallmarks	In Vitro Study	In Vivo Study	Effect(s)	Reference
MED1	+	+	Decreases cell viability, induces cell-cycle arrest, inhibits tumor growth	[266]
TRIM56	+	+	Suppresses cell proliferation and tumor growth	[267]
Wee1, CDK4, CDK7	+	+	Decreases tumor growth	[268]
MST2	+	+	Induces apoptosis and decreases tumor growth	[269]
ERα	+	+	Decreases proliferation and vascularization of solid tumors	[270]
CDK2	+	+	Inhibits cell proliferation and tumor growth	[271]
Bcl2	+	+	Inhibits cell growth and tumor growth	[272]
LEM4	+	+	Decreases cell proliferation and tumor growth	[273]
PDK1	+	-	Decreases cell proliferation	[274]
ATG5, ATG7	+	-	Inhibits autophagy	[275]
ZNF213	+	-	Inhibits cell proliferation	[276]
RBCK1	+	-	Reduces cell proliferation and G2-M arrest	[277]
S6 kinase 1	+	-	Induces cell death via glucose deprivation	[278]
USP15	+	+	Suppresses cell viability, triggers cell-cycle arrest, decreases tumor growth	[279]
AMPKα1/2	+	-	Decreases cell survival	[280]
CDK7	+	-	Decreases cell viability and increases apoptosis	[281]

+, yes; -, no.

**Table 3 cancers-14-05206-t003:** MiRNAs linked with anti-estrogen therapy resistance of ER+ BC.

MiRNA	Tumor Suppressor	Oncogenic	Target	In Vitro/in Vivo Study	Reference
miR-221	-	+	ERα, TIMP3	+/-	[166,302]
miR-222	-	+	ERα	+/-	[166]
miR-192-5p	-	+	ERα, LY6K	+/-	[304]
miR-500-3p	+	-	ERα, LY6K	+/-	[304]
miR-335	-	+	ERα	+/-	[291]
miR-378a-3p	+	-	GOLT1A	+/-	[285]
miR-484	+	-	KLF4	+/-	[293]
miR-22	-	+	FOXP1, HDAC4	+/-	[305]
miR-375	+	-	ATG7, LC3-II	+/-	[294]
miR-214	+	-	UCP2	+/-	[295]
miR-21	-	+	PTEN	+/-	[296]
miR-27a-3p	-	+	PDL1	+/-	[297]
miR-575	-	+	ERα, CDKN1B, BRCA1	+/+	[298]
miR-143	+	-	KRAS	+/+	[299]
miR-9-5p	-	+	ADIPOQ, ESR1	+/+	[303]
miR-18a	-	+	ERα, WNT	+/-	[300]
miR-26a	+	-	E2F7	+/-	[301]
miR-15b	-	+	FOXO1	+/-	[306]
miR-30c	+	-	ERα, HER4	+/-	[180]
miR-205	+	-	RUNX2, AMOT	+/-	[307,308]
miR-200c	+	-	CMYB	+/-	[309]
miR-204	+	-	SOX4	+/-	[310]
miR-491-5p	+	-	JMJD2B	+/-	[311]
miR-206	+	-	ERα	+/-	[312]
miR-124	+	-	AKT2	+/+	[292]
miR-432-5p	-	+	CDK6	+/-	[313]
miR-223	+	-	E2F1	+/+	[314]
miR-339-5p	+	-	CDK2	+/-	[315]
miR-106b	-	+	p21 and PTEN	+/-	[316]
miR-519a	-	+	PTEN	+/-	[317]
miR-574-3p	+	-	CLTC	+/-	[318]
miR-449a	+	-	ADAM22	+/-	[319]
miR-10b	-	+	HDAC4	+/-	[320]

-, no; +, yes.

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
