# Peer review of "Molecular Mechanisms of Anti-Estrogen Therapy Resistance and Novel Targeted Therapies"

_cancers, 2022, doi:10.3390/cancers14215206_

Round 1

Reviewer 1 Report

The manuscript titled “Molecular Mechanims of Anti-estrogen Therapy Resistance and Novel Targeted Therapies” by Ozyurt et al covers the important aspects of resistance to anti-estrogen therapy and the novel therapies that can be used.  It is an interesting and well-written manuscript but the authors should incorporate the following suggestions to further enhance the quality of their manuscript.

Major revisions

1.      The authors need to emphasize the novelty of their review as there are a few recently published reviews on the same topic. Some of them are mentioned below and are also cited in the manuscript.

·         AlFakeeh, A. & Brezden-Masley, C. Overcoming Endocrine Resistance in Hormone Receptor–Positive Breast Cancer. Current Oncology 25, 18–27 (2018).

·         Belachew, E. B. & Sewasew, D. T. Molecular Mechanisms of Endocrine Resistance in Estrogen-Receptor-Positive Breast Cancer. Frontiers in Endocrinology 12, (2021).

·         Pepermans, R. A. & Prossnitz, E. R. ERα-targeted endocrine therapy, resistance and the role of GPER. Steroids 152, 108493 (2019).

·         Rani, A., Stebbing, J., Giamas, G. & Murphy, J. Endocrine Resistance in Hormone Receptor Positive Breast Cancer–From Mechanism to Therapy. Frontiers in Endocrinology 10, (2019).

·         Szostakowska, M., TrÄ™biÅ„ska-Stryjewska, A., Grzybowska, E. A. & Fabisiewicz, A. Resistance to endocrine therapy in breast cancer: molecular mechanisms and future goals. Breast Cancer Res Treat 173, 489–497 (2019).

2.      Since the manuscript is discussing novel targeted therapies, perhaps they should include a table quoting ongoing/completed clinical trials/or the recent FDA approved drugs in this area. They have mentioned it briefly in the text but a separate table would be better

3.      The authors mention the importance of cancer stem cells in resistance, briefly. Since there is a lot of interest in this area, if possible, they should elaborate a bit on it and also how the novel strategies of overcoming resistance are being designed to target cancer stem cells

4.      The authors talk about the role of miRNAs in anti estrogen therapy resistance but make no mention of lncRNAs. There are many reports which discuss the role of lncRNA in this aspect and it would be apt to add a section on lncRNAs too. A few of these reports are mentioned below:

·         Ma, T. et al. LncRNA LINP1 confers tamoxifen resistance and negatively regulated by ER signaling in breast cancer. Cellular Signalling 68, 109536 (2020).

·         Basak, P. et al. Long Non-Coding RNA H19 Acts as an Estrogen Receptor Modulator that is Required for Endocrine Therapy Resistance in ER+ Breast Cancer Cells. CPB 51, 1518–1532 (2018).

·         Ren, L. et al. Long non-coding RNA FOXD3 antisense RNA 1 augments anti-estrogen resistance in breast cancer cells through the microRNA-363/ trefoil factor 1/ phosphatidylinositol 3-kinase/protein kinase B axis. Bioengineered 12, 5266–5278 (2021).

Minor revisions

1.      In Fig 1, they should mention in the caption what does DBD and LBD stand for. And should also include the molecular weight like they have included in Fig 2

2.      The figure caption for Fig 3 looks incomplete, the authors should briefly describe the role of COR, CoA, AP-1. Also correct the spelling of “polymerase” in the figure

3.      In Fig 4, instead of microRNA, non-coding RNA dysregulation should be written to reflect both miRNAs and lncRNAs

4.      Figure 5 seems to have the same information that is given in Table 2. So, maybe they can remove Fig 5. If they decide to retain Fig 5, the authors need to mention in the figure captions that the genes mentioned next to miRNAs are their direct/indirect targets

5.      Please correct this line “Anti-estrogen therapy-resistant BC cells with ERa loss retain their proliferative advantage R-modulating therapy such as that with tamoxifen” mentioned in Section 2.2

Author Response

Dear editor,

Thank you for the opportunity to revise our review article entitled " Molecular Mechanims of Anti-Estrogen Therapy Resistance and Novel Targeted Therapies".

We addressed all issues raised by the reviewers and uploaded our point by point response.

Sincerely

Bulent Ozpolat, M.D., Ph.D.

Professor

Houston Methodist Research Institute

Department of Nanomedicine

Dr. Marr and Roy Neil Cancer Center

6670 Bertner Avenue

Houston, TX 77030

E-mail: bozpolat@houstonmethodist.org       

Major revisions

  1. The authors need to emphasize the novelty of their review as there are a few recently published reviews on the same topic. Some of them are mentioned below and are also cited in the manuscript.

Response: Our current review summarizes important recent developments in the field, such as the role of non-coding RNAs ( i.e, miRNAs and lncRNAs)  and RNA-based therapies that are  not emhasized in the recent reviews. We also included a summary of updated about the results of recently completed and ongoing clinical trials regarding new oral SERDs, CDK4/6 and PI3K/AKT/mTOR inhibitors and their combinations.

  1. 2. Since the manuscript is discussing novel targeted therapies, perhaps they should include a table quoting ongoing/completed clinical trials/or the recent FDA approved drugs in this area. They have mentioned it briefly in the text but a separate table would be better

Response: In line with your suggestion, we included  a table (Table 1 in revised manuscript), summarizing ongoing, completed and accepted clinical trials or the recent FDA approved drugs for patients with anti-estrogen therapy resistant ER+ breast cancer.

  1. The authors mention the importance of cancer stem cells in resistance, briefly. Since there is a lot of interest in this area, if possible, they should elaborate a bit on it and also how the novel strategies of overcoming resistance are being designed to target cancer stem cells

Response: We included a new subtitle (7.8. Breast cancer stem cells (BCSCs)) further indicating the importance of breast cancer stem cells in anti-estrogen therapy resistance.

  1. The authors talk about the role of miRNAs in anti estrogen therapy resistance but make no mention of lncRNAs. There are many reports which discuss the role of lncRNA in this aspect and it would be apt to add a section on lncRNAs too. A few of these reports are mentioned below:

Response:  We included as  a new subtitle (7.10. Dysregulation of LncRNAs) and elaborated the role of lncRNAs in anti-estrogen therapy resistance.

Minor points:

  1. In Fig 1, they should mention in the caption what does DBD and LBD stand for. And should also include the molecular weight like they have included in Fig 2

Answer: Figure 1 has been modified according to your suggestion.

  1. The figure caption for Fig 3 looks incomplete, the authors should briefly describe the role of COR, CoA, AP-1. Also correct the spelling of “polymerase” in the figure

Answer: Figure 3 has been redrawn. We included an explanation for this part and corrected the spelling error.

  1. In Fig 4, instead of microRNA, non-coding RNA dysregulation should be written to reflect both miRNAs and lncRNAs

Answer: Figure 4 was modified as suggested.

  1. Figure 5 seems to have the same information that is given in Table 2. So, maybe they can remove Fig 5. If they decide to retain Fig 5, the authors need to mention in the figure captions that the genes mentioned next to miRNAs are their direct/indirect targets

Answer: We removed the Figure 5 as suggested.

  1. Please correct this line “Anti-estrogen therapy-resistant BC cells with ERa loss retain their proliferative advantage R-modulating therapy such as that with tamoxifen” mentioned in Section 2.2

Answer: This sentence was removed.

Reviewer 2 Report

This is a good review of the actual situation of mechanisms of resistance to endocrine therapy in ER+ brest cancer.

Simple summary: Lethality is driven by endocrine resistance- not only – maybe I would stateis driven by endocrine resistance “amongst others”.

In page 2. HER2-enriched is not an immunhitochemical subtype, I would say HER2 positive instead.

The sentence accounting for most BC deaths in page 2 first paragraph -> I would substitute it with driving or leading to.

The paragraph introducing part 1.1 is too long: In this review, we present the current status of anti-estrogen therapy resistance, existing and more therapeutic opportunities; furthermore, we evaluate potential therapies that need to be considered to further enhance strategies for overcoming anti-estrogen resistance.

I would rather say: “In this review, we present the current status of anti-estrogen therapy resistance, the existing options and further potential therapies that need to be considered to enhance strategies for overcoming anti-estrogen resistance”.

In part 1, I agree that 1.4, 1.5 and 1.6 are different endocrine therapy treatments but I would estructure 1.1, 1.3 and 1.4 as pathways involved in ER signalling and not as part of the existing strategy treatments. Maybe it would be good to separate them into two different parts.

There is an error on the title 1.6- Aromatize should be aromatase.

In part 2 authors mention the most relevant mechanisms of resistance to AI known to date. In gerenal they mention the most important ones, however, authors are missing one big issue amongst others: HER2 status and the interaction between HER 2 E and ER. I Would recommend them to read and cite:

1.     Bergamino MA, López-Knowles E, Morani G, Tovey H, Kilburn L, Schuster EF, Alataki A, Hills M, Xiao H, Holcombe C, Skene A, Robertson JF, Smith IE, Bliss JM, Dowsett M, Cheang MCU; POETIC investigators. HER2-enriched subtype and novel molecular subgroups drive aromatase inhibitor resistance and an increased risk of relapse in early ER+/HER2+ breast cancer. EBioMedicine. 2022 Sep;83:104205. doi: 10.1016/j.ebiom.2022.104205. Epub 2022 Aug 16. PMID: 35985932; PMCID: PMC9482930.

In addition,they are also missing some essential papers on mechanisms of resistance to AI in ER+ breast cancer from Matt Ellis group, who has led the research on this field worldwide. They should also include and investigate some of the mechanisms included in these papers.

2.     Ma CX, Reinert T, Chmielewska I, Ellis MJ. Mechanisms of aromatase inhibitor resistance. Nat Rev Cancer. 2015 May;15(5):261-75. doi: 10.1038/nrc3920. PMID: 25907219.

3.     Anurag M, Zhu M, Huang C, Vasaikar S, Wang J, Hoog J, Burugu S, Gao D, Suman V, Zhang XH, Zhang B, Nielsen T, Ellis MJ. Immune Checkpoint Profiles in Luminal B Breast Cancer (Alliance). J Natl Cancer Inst. 2020 Jul 1;112(7):737-746. doi: 10.1093/jnci/djz213. PMID: 31665365; PMCID: PMC7805027.

I would change title of part 3 – Current and potential treatments to improve efficacy of current anti-endrogen treatments / or to minimize resistance to endocrine therapy.

Please review that you do not write acronyms in the titles, for example BC in part 3 title.

In 3.2 – the majority would not be precise, at around 50% of cases.

In the same first paragraph of section 3.2 – to overcome ER+ BC resistance is missing.

Author Response

Dear editor,

Thank you for the opportunity to revise our review article entitled " Molecular Mechanims of Anti-Estrogen Therapy Resistance and Novel Targeted Therapies".

We addressed all issues raised by the reviewers and uploaded our point by point response.

Sincerely

Bulent Ozpolat, M.D., Ph.D.

Professor

Houston Methodist Research Institute

Department of Nanomedicine

Dr. Marr and Roy Neil Cancer Center

6670 Bertner Avenue

Houston, TX 77030

E-mail: bozpolat@houstonmethodist.org       

  1. Simple summary: Lethality is driven by endocrine resistance- not only – maybe I would stateis driven by endocrine resistance “amongst others”.

Answer: We revised the sentence as suggested by the reviwer.

  1. In page 2. HER2-enriched is not an immunhitochemical subtype, I would say HER2 positive instead.

Answer: We corrected the sentence as suggested.

  1. The sentence accounting for most BC deaths in page 2 first paragraph -> I would substitute it with driving or leading to.

Answer: We substituted it with 'driving' as suggested.

  1. The paragraph introducing part 1.1 is too long: In this review, we present the current status of anti-estrogen therapy resistance, existing and more therapeutic opportunities; furthermore, we evaluate potential therapies that need to be considered to further enhance strategies for overcoming anti-estrogen resistance. I would rather say: “In this review, we present the current status of anti-estrogen therapy resistance, the existing options and further potential therapies that need to be considered to enhance strategies for overcoming anti-estrogen resistance”.

Answer: The paragraph was revised as suggested.

  1. In part 1, I agree that 1.4, 1.5 and 1.6 are different endocrine therapy treatments but I would estructure 1.1, 1.3 and 1.4 as pathways involved in ER signalling and not as part of the existing strategy treatments. Maybe it would be good to separate them into two different parts.

Answer: We rearranged the subtitles.

  1. There is an error on the title 1.6- Aromatize should be aromatase.

Answer: It was corrected as ‘aromatase’.

  1. In part 2 authors mention the most relevant mechanisms of resistance to AI known to date. In gerenal they mention the most important ones, however, authors are missing one big issue amongst others: HER2 status and the interaction between HER 2 E and ER. I Would recommend them to read and cite “Bergamino MA etal

Answer: Thank you for your suggestion. We cited it in the manuscript.

  1. In addition,they are also missing some essential papers on mechanisms of resistance to AI in ER+ breast cancer from Matt Ellis group, who has led the research on this field worldwide. They should also include and investigate some of the mechanisms included in these papers.

Answer:  We included mechanims that were mentioned in these papers.

  1. I would change title of part 3 – Current and potential treatments to improve efficacy of current anti-endrogen treatments / or to minimize resistance to endocrine therapy.

Answer: The title (part 6) was changed as suggested. Part 3 corresponds to part 6 of the revised manuscript as the subtitles have been rearranged.

  1. Please review that you do not write s in the titles, for example BC in part 3 title.

Answer: Titles were edited as  suggested.

  1. In 3.2 – the majority would not be precise, at around 50% of cases.

Answer: Part 3.2 corresponds to part 6.2 of the revised manuscript as the subtitles have been rearranged. The sentence was corrected as ysuggested.

  1. In the same first paragraph of section 3.2 – to overcome ER+ BC resistance is missing.

Answer: Missing word was added (PI3K/AKT/mTOR signaling is activated in advanced ER+ BC; thus, this signaling is proposed to be an important target to overcome ER+ BC resistance).

Reviewer 3 Report

The manuscript entitled ‘Molecular Mechanims of Anti-estrogen Therapy Resistance and Novel Targeted Therapies’ provided a comprehensive review about the molecular mechanism and drug resistance of ER positive breast cancer. The suggestions are addressed below. 1. Please do not subtitle the first section ‘Introduction’. Please focus on the purpose of this review. ‘1.1. to 1.6.’ should be list in section two for another subtitle, such as ‘the mechanism and regulation of estrogen signal pathway’. 2. Please describe the ncRNA role in mechanism of resistance to anti-estrogen therapies, beside miRNA. 3. The figures are unclear. Please re-draw all of the figures. The size of words and lines are unsuitable. All abbreviation should be described in the figure legends. 5. If the authors list the current research about the potential targets for resistance situation, the title is not suitable, as ‘Novel Targets Therapies’. Please revise the title for specific purpose. 6. The table did not acceptable. It is suggested to classify as the cancer hallmarks, instead of the name of proteins. Otherwise too confused.

Author Response

Dear Reviewer,

Thank you for the opportunity to revise our review article entitled " Molecular Mechanims of Anti-Estrogen Therapy Resistance and Novel Targeted Therapies".

We addressed all issues raised by the reviewers and uploaded our point by point response.

Sincerely

Bulent Ozpolat, M.D., Ph.D.

Professor

Houston Methodist Research Institute

Department of Nanomedicine

Dr. Marr and Roy Neil Cancer Center

6670 Bertner Avenue

Houston, TX 77030

E-mail: bozpolat@houstonmethodist.org       

  1. Please do not subtitle the first section ‘Introduction’. Please focus on the purpose of this review. ‘1.1. to 1.6.’ should be list in section two for another subtitle, such as ‘the mechanism and regulation of estrogen signal pathway’.

Answer: We removed the subtitle from the introduction part. All subtitles were rearranged as you suggested.

  1. Please describe the ncRNA role in mechanism of resistance to anti-estrogen therapies, beside miRNA.

Answer: We added the long noncoding RNAs  part (Subtitle 5.10. Dysregulation of long noncoding RNAs (LncRNAs)).

  1. 3. The figures are unclear. Please re-draw all of the figures. The size of words and lines are unsuitable.

Answer: Figure 1 and figure 4 have been modified. Figure 5 is removed. Figure 3 has been re-drawn.

  1. All abbreviation should be described in the figure legends.

Answer: All abbreviations were corected.

  1. The table did not acceptable. It is suggested to classify as the cancer hallmarks, instead of the name of proteins. Otherwise too confused.

Answer: Instead of the name of the proteins, cancer hallmarks were written in table.

Round 2

Reviewer 1 Report

The authors have answered all the comments. However there are minor spelling mistakes and grammatical errors. I would suggest that the authors proofread the manuscript once again.